# Ligand-specific conformational change drives interdomain allostery in Pin1

Alexandra Born [1], Janne Soetbeer [2], Morkos A. Henen[1,3], Frauke Breitgoff[2], Yevhen Polyhach[2], Gunnar Jeschke [2] & Beat Vögeli [1] ✉

Pin1 is a two-domain cell regulator that isomerizes peptidyl-prolines. The catalytic domain (PPIase) and the other ligand-binding domain (WW) sample extended and compact conformations. Ligand binding changes the equilibrium of the interdomain conformations, but the conformational changes that lead to the altered domain sampling were unknown. Prior evidence has supported an interdomain allosteric mechanism. We recently introduced a magnetic resonance-based protocol that allowed us to determine the coupling of intra- and interdomain structural sampling in apo Pin1. Here, we describe ligand-specific conformational changes that occur upon binding of pCDC25c and FFpSPR. pCDC25c binding doubles the population of the extended states compared to the virtually identical populations of the apo and FFpSPR-bound forms. pCDC25c binding to the WW domain triggers conformational changes to propagate via the interdomain interface to the catalytic site, while FFpSPR binding displaces a helix in the PPIase that leads to repositioning of the PPIase catalytic loop.

Many proteins are organized into multiple domains with the potential for interdomain crosstalk. The function and activity of one domain can be modulated through the structure and dynamics of another domain, what we thus term "interdomain allostery". In addition, the structure and dynamics of the entire system may change upon environmental change (i.e., substrate binding). While interdomain allostery can be demonstrated by comparing the activity of the individual domains to the whole multi-domain system, it is challenging to directly probe the cascades of conformational changes leading to allostery.

Extensive work has focused on investigating interdomain allostery through the two-domain mitotic regulator Pin1[1–7]. Pin1 regulates other proteins through isomerization of the peptide bonds of prolines that leads to the recycling of phosphoproteins, regulating other post-translational modification, and enhancing or preventing protein degradation[8]. Pin1 consists of the WW binding domain (residues 1–39) connected to the peptidyl-prolyl isomerase (PPIase) domain (residues 50–163) via a flexible linker (Fig. 1a and Supplementary Fig. 1b). Pin1's PPIase domain isomerizes the peptide bonds of prolines that are

immediately preceded by either a phosphorylated serine or threonine (pS/T-P motif). While the PPIase can catalyze both *cis-trans* or *trans-cis* reactions, the WW domain binds *trans*-specifically. The domains have been shown to populate both a compact state with a specific interdomain (ID) interface as well as a dispersed extended state where the two domains can tumble semi-independently as displayed in Fig. 1b, c. Our recent work suggests that apo Pin1 occupies a compact and extended state in a 70:30 ratio[9].

The primary ligands that have been biophysically evaluated are FFpSPR (peptide sequence) and pCDC25c (with peptide sequence EQPLpTPVTDL). FFpSPR is a more soluble variant of another artificial ligand (Pintide with peptide sequence WFYpSPR) that has high PPIase efficiency with minimal sequence[10]. The pCDC25c ligand is a peptide derivative from the pT48-P49 site of the mitotic phosphatase CDC25c[11]. An early study of interdomain mobility compared tumbling times ($\tau_c$) of apo Pin1 and Pin1 complexed to various ligands. The study concluded that Pintide restricts the flexibility of the two domains (i.e., induces a more compact state), while pCDC25c increases the

[1]University of Colorado Anschutz Medical Campus, Department of Biochemistry and Molecular Genetics, Aurora, CO, USA. [2]Laboratory of Physical Chemistry, Department of Chemistry and Applied Biosciences, ETH Zürich, Vladimir-Prelog-Weg 2, ETH-Hönggerberg, Zürich, Switzerland. [3]Faculty of Pharmacy, Mansoura University, Mansoura, Egypt. ✉e-mail: beat.vogeli@cuanschutz.edu

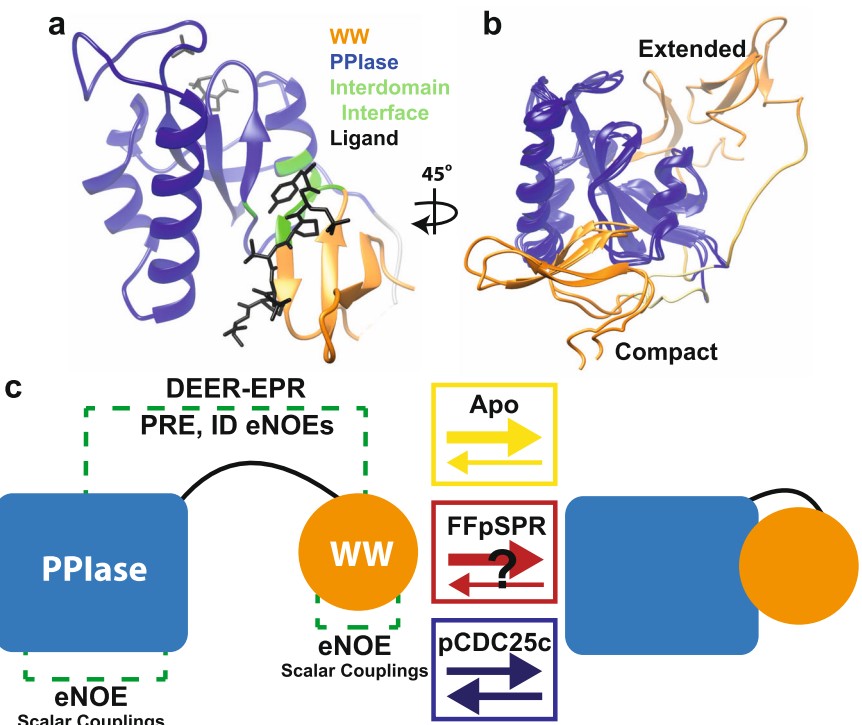

**Fig. 1 | Multiple conformations in Pin1. a** Crystal structure 1PIN[17] overlaid with C-terminal domain peptide from crystal structure 1F8A[11]. PPIase domain, WW domain, interdomain interface and ligand are shown in blue, orange, green and black, respectively. **b** Two-state, two-conformer NMR ensemble showing compact and extended states in apo Pin1. **c** Schematic of conformational changes due to ligand binding in Pin1 and experimental methods used to detect them.

flexibility[7]. Further NMR studies involving chemical shift perturbations (CSPs) and order parameters quantifying the motional restriction of covalent bonds have supported the reduced interdomain contact caused by pCDC25c[1,2,5,12], but the proposed compaction due to FFpSPR binding is not supported by published data[6] (Fig. 1c). As the divergent CSPs mainly occur on the interdomain interface (Supplementary Fig. 1), they indeed indicate that these ligands cause distinct allosteric effects and interdomain orientations.

It has been established that the WW domain can allosterically regulate the activity of the PPIase[1,2]. While only the PPIase domain is catalytically active, mutagenesis studies have shown that the WW domain is essential in vivo[13]. Yet, the isolated PPIase domain displays higher catalytic activity than the full-length protein for both FFpSPR and pCDC25c ligands[2]. Therefore, WW domain contact alters the properties of the PPIase catalytic site, which is located on the opposite side of the PPIase domain than the interface with the WW domain. Overall, these observations support interdomain allostery coupled to intradomain allostery. Interdomain communication and dynamic allostery are also supported by the fact that ligand-binding reduces methyl group flexibility along a conduit linking the ID interface to the catalytic site[5]. The exact conduit residues and degree of change shows some variance depending on the ligand and its effect on the interdomain contact[2,6]. More evidence of interdomain allostery comes from the expansive studies on the I28A mutation in the WW domain interface. This mutation in the WW domain increases the activity of the PPIase with evidence supporting the stabilization of the extended state[1,12,14]. A conformational selection-driven allosteric model has been proposed that explains why PPIase activity is negatively regulated by interdomain contact[1,2]. Both extended and compact states exist in apo Pin1; shifting the equilibrium to favor the extended state by either mutating the interface, binding to pCDC25c, or through deleting the WW domain causes an increase in catalysis[1]. In addition, a recent study shows that pCDC25c binding causes compaction within the WW domain while the interdomain distance increases[15]. In contrast, other ligands, including FFpSPR, do not have the same effect on the interdomain contact[4,16]. All this data establishes that ligand binding changes the intradomain structure and interdomain orientation, and that the specific ligand sequence has distinct effects on Pin1.

An extensive molecular dynamics study suggests that two pathways mediate the interdomain allosteric regulation of ligand FFpSPR: Path 1 emanates from the canonical interdomain interface and leads to the PPIase β-sheet core, α4, and the loop containing residues 152–154, and Path 2 starts at the α1-α2 loop (residues 98–102), α2, and the catalytic loop[4]. Upon addition of FFpSPR, Path 2 extends into the WW domain's ligand-binding pocket via the bound substrate[4]. In the following, we adapt this nomenclature (see Fig. 5).

Previous work has extensively described signatures of allostery in Pin1, yet to date the structure of ligand-bound, full-length Pin1 in solution has not been determined. Though ligand-bound crystal structures exist[17], they show Pin1 only occupying a compact configuration. We recently published a protocol for elucidating the coupling of inter- and intradomain spatial sampling of multi-domain proteins using magnetic resonance[9]: first, we calculated multi-state structural ensembles of apo Pin1 from exact Nuclear Overhauser Enhancements (eNOE) and scalar couplings restraining the intradomain structure, whereas paramagnetic relaxation enhancement (PRE), interdomain NOEs and residue dipolar couplings (RDC) define the domains' positions. In a second step, our ensemble was cross-validated against experimental double electron-electron resonance (DEER) electron paramagnetic resonance (EPR) measurements. Our two-state ensemble (PDB ID: 7SA5) simultaneously describes the compact and extended states present in solution and shows coupling between intradomain conformation and interdomain positions through rearrangement of hydrophobic residues in the interdomain interface extending to the catalytic pocket[9]. Here, we apply the same method[9] as summarized in Fig. 1c to determine the intra- and interdomain conformational changes due to ligand binding by solving the structure of Pin1 in presence of saturating amounts of pCDC25c and FFpSPR. We

obtain ensembles that aid in describing the structural rearrangements upon ligand binding and form the basis of the allosteric mechanisms.

## Results and discussion

### pCDC25c, but not FFpSPR, stabilizes the extended state

In agreement with previous reports of Pin1[2,7], we observe that upon pCDC25c addition the domain-specific tumbling time ($\tau_c$) of the WW domain decreases relative to that of the PPIase domain (but not FFpSPR). As shown in Supplementary Table 1, this indicates decoupling of the two domains and stabilization of extended states. Moreover, we observe between 20 and 26 interdomain NOEs in our spectra of Pin1 with near saturating amount of FFpSPR and pCDC25c, which have not been reported in previous studies, supporting the partial sampling of compact states.

We utilize DEER to directly measure the distance distribution between the two domains. These EPR measurements involved flash-freezing the samples, allowing for the detection of all distances and their populations at the temperature where the sample vitrifies. Using the double-mutant MTSL-labeled constructs 15–90, 15–98, and 15–131, we measured distances between the two domains for apo and ligand-bound Pin1, while construct 90–131 with both labels in the PPIase domain served as a control (Fig. 2a). We performed 4- or 5-pulse DEER measurements and various analyses to detect distances between 15 and 80 Å (Supplementary Fig. 2, detailed analysis in Supplementary Results and Discussion section). As expected, the control mutant 90–131 shows narrow distributions that remain nearly unchanged upon ligand addition. While all measurements were fit with unparametrized approaches, constructs 15–90 and 15–98 could also be well described using a bi-Gaussian model (Fig. 2a).

Regardless of ligand presence or absence, for constructs 15–90 and 15–98, we see a narrow distance distribution centered around short distances of 22 and 24 Å, respectively, corresponding to a compact conformation (Fig. 2a, b). A longer, dispersed distance is also sampled and centered around 45 Å, corresponding to the extended

state. We have shown previously that apo Pin1 occupies a compact configuration to ~70%[9]. Our measurements show that ligand FFpSPR does not considerably alter this population of the compact state ($p_1$) as it changes by less than ±5% for both 15–90 and 15–98 constructs compared to apo Pin1 (Fig. 2b). Conversely, we see clear proof that pCDC25c stabilizes the extended state: for construct 15–90 the population of the compact state decreases from 68% to 50%, whereas for 15–98 the compact population only decreases by 7%. We attribute this population discrepancy to the significant broadening of the extended conformation in the 15–98 construct (Fig. 2a, Supplementary Tables 2 and 3, $\sigma_2$ parameters), and the wider rotamer distribution of 98 compared to 90 as previously described[9], both leading to a higher uncertainty in the population for 15–98. Similarly, with PRE measurements (Supplementary Fig. 3), we see a decrease in enhancement of the transverse relaxation rate using MTSL-labeled samples ($R_2^{sp}$) across domains upon addition of pCDC25c (but not FFpSPR), which also corresponds to an increased distance between the domains. From $R_2^{sp}$, we extracted population-averaged interdomain distances (up to 25 Å) that are used as distance restraints in our subsequent structure calculations.

For the 15–131 construct, the bi-Gaussian model provides a poor description of the distance distribution. For this reason, we are unable to provide a quantitative $p_1$ value. Nevertheless, we observe two major populations centered around 38 and 43 Å. For apo and FFpSPR-bound Pin1, it appears that the 43 Å peak is populated twice as often as the 38 Å peak, which is a similar ratio (66:33) that we see for the compact and extended populations with the other DEER constructs. This population near 43 Å decreases upon addition of pCDC25c. Due to the correlation of populations between the 15–90 and 15–131 constructs, we believe that the peaks centered around 43 and 38 Å originate from the compact and extended states, respectively. Upon pCDC25c addition, we also see an increase in smaller distances between 20 and 35 Å and at larger distances around 55 Å that may balance the decrease in intensity around 43 Å. Based on our DEER measurements, we conclude

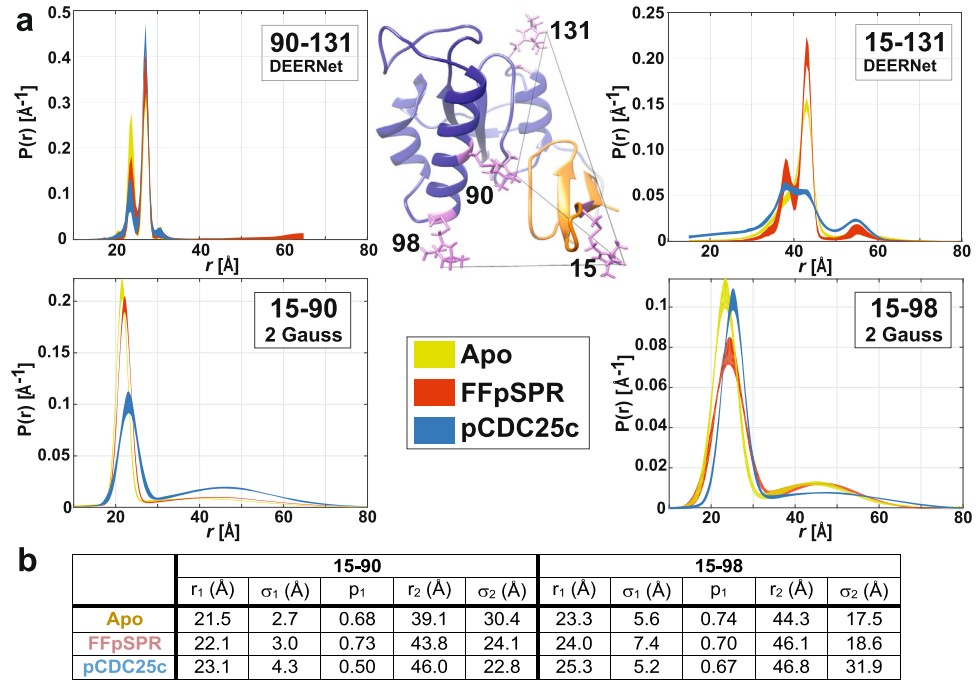

**b**

| | **15-90** | | | | | **15-98** | | | | |
|---|---|---|---|---|---|---|---|---|---|---|
| | $r_1$ (Å) | $\sigma_1$ (Å) | $p_1$ | $r_2$ (Å) | $\sigma_2$ (Å) | $r_1$ (Å) | $\sigma_1$ (Å) | $p_1$ | $r_2$ (Å) | $\sigma_2$ (Å) |
| **Apo** | 21.5 | 2.7 | 0.68 | 39.1 | 30.4 | 23.3 | 5.6 | 0.74 | 44.3 | 17.5 |
| **FFpSPR** | 22.1 | 3.0 | 0.73 | 43.8 | 24.1 | 24.0 | 7.4 | 0.70 | 46.1 | 18.6 |
| **pCDC25c** | 23.1 | 4.3 | 0.50 | 46.0 | 22.8 | 25.3 | 5.2 | 0.67 | 46.8 | 31.9 |

**Fig. 2 | Impact of ligand binding on the DEER distance distributions of Pin1.**
**a** DEER distributions are plotted including 95% confidence intervals for each construct obtained from either a DEERNet neural network analysis or bi-Gaussian fit. Apo, FFpSPR- and pCDC25c-bound Pin1 data is plotted in yellow, red and blue, respectively. MTSL positions (pink) and distances (black) are drawn onto the WW (orange) and PPIase (blue) domains of the x-ray structure 1pin[17]. **b** Gaussian parameters for 15–90 and 15–98 constructs, where $r_i$ = distance, $\sigma_i$ = distance standard deviation and $p_i$ = population with index $i = 1$ and $i = 2$ corresponding to the compact and extended state, respectively. See Supplementary Tables 2–4 for 95% confidence intervals, which are typically ±0.03 for populations.

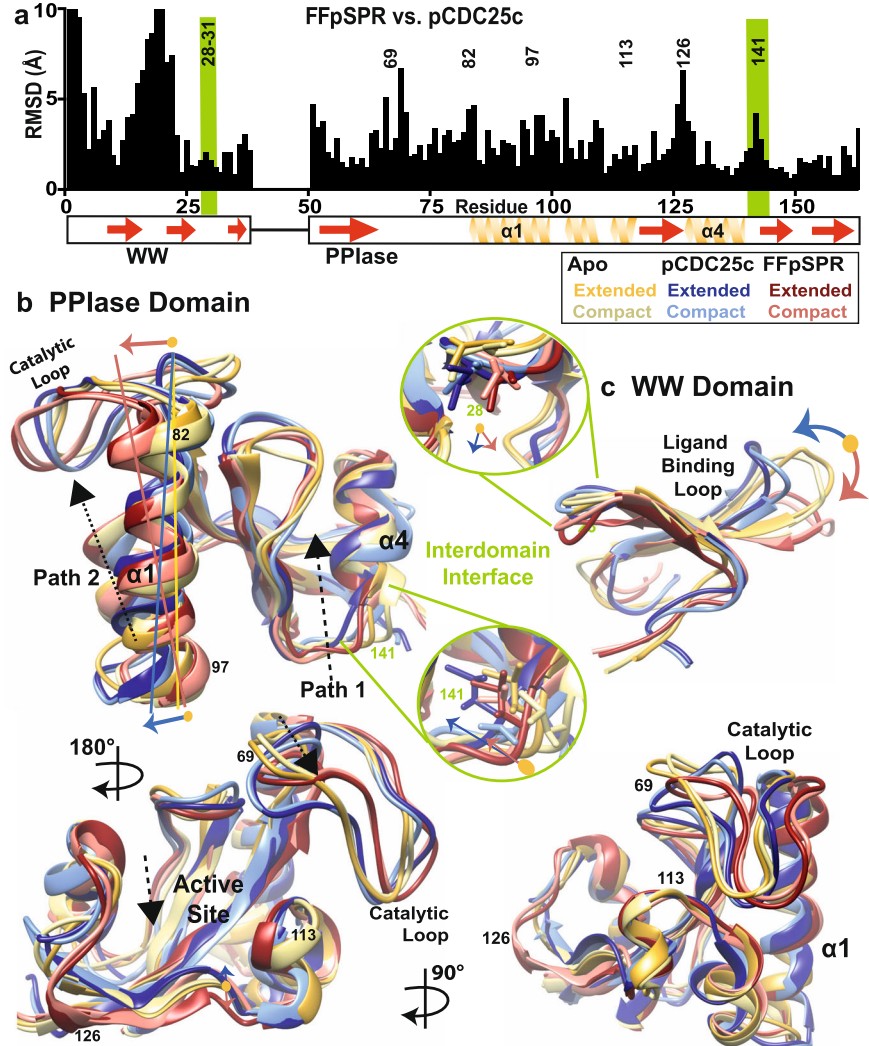

**Fig. 3 | Comparison of two-state structures of apo, pCDC25c-bound, and FFpSPR-bound Pin1. a** RMS deviation (including side-chain) between the mean FFpSPR- and pCDC25c-bound structures (all conformers) is plotted versus residue number. Key residues are noted, and interdomain interface is shaded green. Superposition of mean two-state structures of the **b** PPIase and **c** WW domain. Extended/compact states of apo, FFpSPR- and pCDC25c-bound Pin1 are drawn in dark yellow/light yellow, dark blue/light blue and maroon/pink.

that FFpSPR-binding does not significantly change the apo equilibrium (70:30), while pCDC25c increases the extended population between 7 and 20%.

Independent qualitative assessment of these population shifts comes from slow-exchange NMR peaks observed for residues located in the interdomain interface (Supplementary Fig. 3e–g)[3,9,18,19]. As multiple exchange peaks exist, we are unable to conclusively assign specific conformations to specific exchange peaks. Nevertheless, the populations of the compact and extended states from the 15–90 and 15–98 DEER distance distributions are similar to the major and most separate minor slow-exchange peak populations from the N90C (~0.66:0.34) and S98C PRE mutant constructs (~0.78:0.22). Thus, we suggest that these major and minor peaks emanate predominantly from the compact and extended conformations, respectively. This is further supported by the absence of interdomain NOESY peaks for the minor exchange peaks, as expected for extended states. Because the extended state does not contribute appreciably to the interdomain NOEs, we cannot discount the possibility that though the major peaks stem primarily from the compact state also the extended state contributes. Furthermore, we also cannot discount that the relevant timescales for interdomain mobility also cover fast exchange which does not result in separate exchange peaks. Therefore, the most

separate slow-exchange minor peaks set a lower limit to the extended population. We note that the relative intensities of these peaks also appear to be sensitive to mutations. In apo and FFpSPR-bound WT Pin1, the average population of the minor peaks is about 10%. Importantly, we consistently see increases in the minor, extended state population upon addition of pCDC25c in both the PRE mutants and WT Pin1 (at least 10% and 5%, respectively). While the populations must be interpreted with caution, we can confidently conclude that the NMR data confirms the DEER data in that pCDC25c, but not FFpSPR, increases the population of the extended state.

## NMR restraints indicate global structural changes upon ligand binding

In order to deduce the inter- and intradomain conformational changes induced by ligand binding, we calculated the two-state NMR ensemble using eNOEs, scalar couplings, PRE and DEER restraints as previously described[9]. As for apo Pin1[9], a two-state ensemble resulting in compact and extended states is necessary for both ligand-bound complexes to satisfy all the experimental data (see Methods for details on the structure calculation and Supplementary Table 5 for structural statistics). A simple analysis of the restraints foreshadows the global structural changes that we will obtain for the two-state ensembles (Fig. 3a).

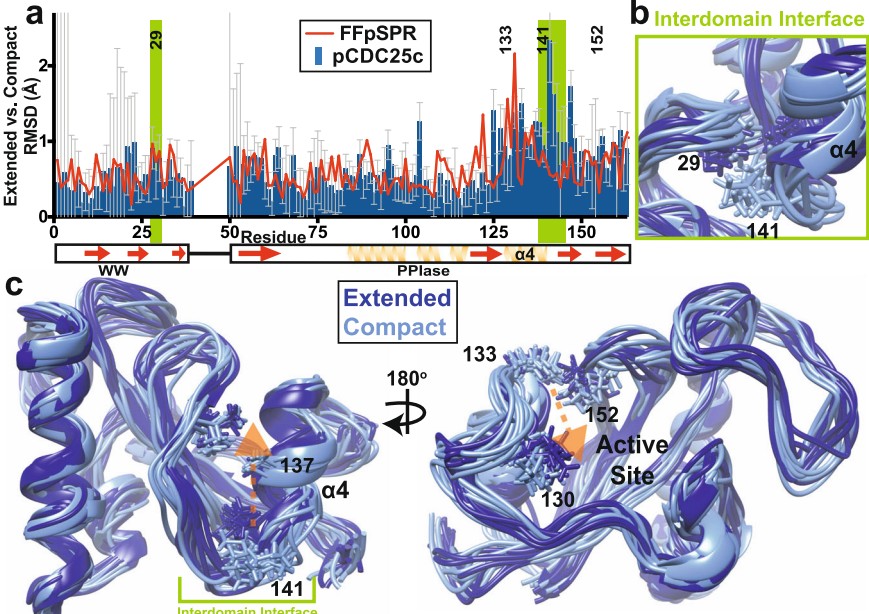

**Fig. 4 | Coupling of inter- and intradomain spatial sampling of pCDC25c-bound Pin1.** The two-state ensemble is analyzed with the states sorted by presence or absence of interdomain contact (compact or extended state, respectively). **a** For the pCDC25c-bound ensemble, RMSD (including side-chain) between mean compact and extended states of domains is plotted in blue versus residue number. The error bars depict the square root of the sum of the squared variances for the compact and extended states alone, providing an upper limit for the uncertainty of the mean difference between the mean compact and extended states. The RMSD versus residue number of the FFpSPR-bound Pin1 two-state ensemble is overlaid in red in the graph. Ten two-state ensembles were used for all calculations. The interdomain interface is highlighted green in the graph. Major RMS deviations between the two states of pCDC25c-bound Pin1 are shown in light and dark blue in the **b** interdomain interface, and **c** PPIase domain. Conformational changes within Path 1 propagate from the interdomain interface to the catalytic site, as shown via the orange arrow.

Comparing the restraint input sets of FFpSPR-, pCDC25c-bound and apo Pin1 to the other two sets shows that 52–63% eNOE, 72–84% PREs, and 60–91% scalar couplings are shared. In part, loss of shared restraints is caused by changes in chemical shifts due to binding (see Supplementary Fig. 1c) that may resolve some overlapped peaks and overlap some resolved peaks. In addition, many NOE cross peaks appear or disappear due to changes in distances, which results in the lowest shared portion among the different restraint types. Supplementary Fig. 4 shows examples of comparisons between NOESY spectra from FFpSPR-, pCDC25c-bound and apo Pin1, and eNOE restraints in the WW domain that are unique to one structure with respect to any of the other two. To determine whether the specific conformational changes in our structural ensembles are only a result of these unique restraints, we also calculated ensembles with identical subsets of restraints. Even with omission of unique restraints, the ligand-dependent conformational changes are still present, but generally with a reduced difference between the states. This indicates that the structural changes are encoded both in shared *and* unique restraints.

## pCDC25c binding induces correlated intradomain conformational changes

Next, we highlight the major conformational changes induced by pCDC25c interaction with Pin1. In the WW domain (Fig. 3a, c and Supplementary Fig. 5b right), the ligand-binding loop (residues 15–22) folds upon pCDC25c binding with a RMSD of up to 8 Å. Previous crystal structures (Fig. 1a) have shown that ligands bind on top (in our view) of this loop and within the pocket[11,17]. This increased compaction within the WW domain upon pCDC25c binding was also reported in a recent study using PRE and MD simulations[15]. The presence of the ligand and the conformational change in this loop appears to also perturb residues 34–36 in the β3 strand Supplementary Fig. 5b right), leading to a tilt which

ultimately also changes the interdomain interface itself. The methyl group of T29 points directly into the interface (or into the PPIase itself) without ligand present, while the methyl group is more excluded from the interface (by pointing downward) when pCDC25c is present. The degree of methyl occlusion from the interface with pCDC25c is likely dependent on the compact and extended states of the two-state ensemble.

Within the interdomain interface (Fig. 3b and Supplementary Fig. 5c right), we see the methyl group of L141 further occluded from solvent in the presence of pCDC25c that results in stabilization of the extended state. The PPIase interdomain interface is located near and within the C-termini of the α4 helix. Upon pCDC25c binding, this helix displays small changes up until its N-termini (i.e., residue P133), and conformational changes also occur in residues 128–131 that make up part of a loop responsible for ligand-binding in the PPIase. We also note clear correlated differences throughout this helix and ligand-binding loop between the compact and extended state of pCDC25c-bound Pin1 (Fig. 5), highlighting distinct motions induced by WW contact. Compared to the apo form, there is also a shift in the core β-sheet propagating from nearby the interdomain interface β6, passing through β7, β4, and finally β5 (Supplementary Fig. 5c bottom right). Residues located in these core β-strands are critical for ligand binding and catalysis in the PPIase. In addition, minor reorientation of the α1 helix propagates conformational changes to the α2 helix and the catalytic loop. The C-terminus of the long α1 helix appears to interact with the WW domain in the compact state. The N-terminus of the α1 helix is connected to the PPIase catalytic loop (residues 65–82), which also is slightly repositioned after binding to pCDC25c. Overall, the PPIase active site appears to exhibit connections to the interdomain interface involve firstly the core β-sheet and C-terminal ligand-binding region (residues 128–131) and secondly the catalytic loop including residues 68 and 69. The latter is responsible for binding the ligand's phosphate.

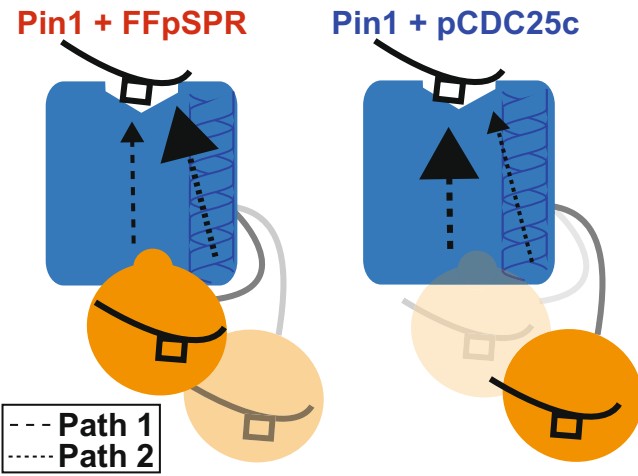

**Pin1 + FFpSPR**    **Pin1 + pCDC25c**

- - - **Path 1**
..... **Path 2**

**Fig. 5 | Proposed model of ligand-dependent conformational changes in Pin1.**
FFpSPR and pCDC25c ligands (black) activate an interdomain allosteric pathway
(Path 1) in Pin1 across the main interface between the WW (orange) and PPIase
domains (blue). Binding of FFpSPR makes use of a second pathway (Path 2) via helix
1 in the PPIase domain. Binding of pCDC25c decreases the population of the
compact state of Pin1, increases the population of the extended state and does not
measurably activate Path 2.

## FFpSPR induces different conformations than pCDC25c

In contrast to pCDC25c-binding, the distance distributions determined
from DEER measurements demonstrate that no major interdomain
rearrangement occurs in presence of ligand FFpSPR. Our two-state
structural ensemble of FFpSPR-bound Pin1 (Supplementary Fig. 5a)
also reveals distinct intradomain conformational changes that differ
from the pCDC25c-bound ensemble (Fig. 3a).

We note many changes in Pin1's structure dependent on the ligand.
Starting with the WW domain: the ligand-binding loop folds upward
upon binding to pCDC25c; conversely it moves in the opposite direc-
tion with FFpSPR (Fig. 3c and Supplementary Fig. 5b left). The
arrangement of the WW domain's interdomain interface is also
dependent on ligand, as key residue I28 moves down and diagonally
with FFpSPR, while pCDC25c only causes the side chain to move down
(Fig. 3c). Additionally, in presence of pCDC25c the side chains of I28
and T29 are more blocked from the interface than with FFpSPR. In the
PPIase domain's interdomain interface, FFpSPR eliminates the corre-
lation between the two states (Fig. 5a). FFpSPR does not increase the
stability of the interdomain extended state. Unlike pCDC25c, FFpSPR
only causes minor changes in the β-sheet core and the α4 helix. How-
ever, we see a major change in the orientation of the α1 helix and the
subsequent rearrangement of the PPIase catalytic loop (residues 65–80)
upon addition of FFpSPR (Fig. 3b and Supplementary Fig. 5c left).

## Domain positions are mediated by spatial sampling at the interdomain interface

Next, we determined the subtle, intradomain conformational changes
that induce the large-scale, interdomain rearrangements. As for apo
Pin1[9], the two-state ligand-bound structural ensembles (Supplemen-
tary Fig. 5) also produce both compact and extended states which were
analyzed to determine correlations between intradomain structure
and interdomain organization. In apo Pin1, we detected methyl rear-
rangement at the PPIase domain's interdomain interface: in the com-
pact state the methyl groups of A140 and L141 point into the
interdomain space, while in the extended state these methyl groups
point back into the PPIase domain itself[9]. The interdomain interface
of the WW domain is also composed of hydrophobic residues, so that this
relatively small conformational change (-2.5 Å) at the interface likely
stabilizes the compact and extended states in apo Pin1 due to a
hydrophobic effect. In addition, the extended and compact states

feature distinct conformations around loops encompassing residues
98–102, 125–128, and 152–154 and the α4 helix[9].

For the pCDC25c-bound form, we can see a clear difference
between extended and compact states on both sides of the interface
(encompassing residues 29 and 141 in the graph in Fig. 4a–c). As was
hinted with the apo Pin1 structure, the compact interface residues
point away from their own domain with the ability to make contacts
now with the other domain. Conversely, in the extended state these
hydrophobic residues are occluded from the interface. However, in the
FFpSPR-bound form (Fig. 4a), this difference between the two states is
only maintained in the WW domain's interface (residues 28–30).

As with apo Pin1, in presence of pCDC25c we also observe distinct
extended and compact conformations of the α4 helix encompassing
residues 132–140 (Fig. 4b). As this helix is within the interdomain
interface, we expect this conformational difference to also be driven
by the presence or absence of the WW domain in the interface. There is
also a difference in structure at the PPIase ligand-binding loop residues
125–131 (Fig. 4c, right). These ligand-binding residues are connected to
the interdomain interface via the previously mentioned α4 helix. We
see a similar RMS deviation between the two states in both apo and
pCDC25c-bound ensembles for residues 125–140[9]. This not only indi-
cates a mode of signal transduction, but also implies how interdomain
contact can lead to changes in the ligand-binding site, which in turn
control the activity of Pin1. We do not see similar correlations with
FFpSPR in this region, likely due to the lack of correlations in the PPIase
domain's interdomain interface. FFpSPR binding appears to allosteri-
cally disrupt this link between the PPIase interdomain interface and
this PPIase ligand-binding loop.

## Ligand-specific conformational changes within the previously proposed allosteric network

Ligands FFpSPR and pCDC25c induce an antagonistic conformational
change in the WW-binding loop that leads to a disparate change in the
interdomain interface likely provoked by residues 34–36 and through
the β3 strand. Residues 28–29 are further occluded from the interface
with pCDC25c, while in presence of FFpSPR these residues are further
reorganized (but less buried than pCDC25c). Subsequently, the PPIase
interdomain residues are also further occluded with pCDC25c, while
the residues adopt an intermediate position with FFpSPR. The
pCDC25c-bound structure shows large conformational changes that
originate from the interdomain interface and propagate to the cata-
lytic site through the core β-sheet and α4 helix, in the previously
proposed Path 1 from MD simulations[4]. While FFpSPR also causes some
changes through the interdomain interface and Path 1, our structures
show larger conformational changes propagating from the α1 helix
into the catalytic loop in Path 2. Conversely, pCDC25c does not cause
as substantial change through the α1 helix. We speculate that the dif-
ferent interdomain distance present in the two pathways determines
the preference for changes within Path 1 and 2 (Fig. 5). FFpSPR does not
alter the interdomain distance which allows the ligand to interact
through the α1 helix when bound to the WW domain. On the contrary,
pCDC25c stabilizes the extended state, greatly reducing the ability of
the ligand bound to the WW domain to also interact with the α1 helix.

## Potential biological role of coexistence of extended and compact states

Inter- and intradomain spatial sampling may be coupled in many multi-
domain proteins. We have used our recently introduced NMR- and
EPR-based approach to demonstrate such coupling in the two-domain
protein Pin1, which rationalizes ligand-specific interdomain allostery.
Specifically, FFpSPR binding does not change Pin1's interdomain
equilibrium while pCDC25c stabilizes the extended state. We have
shown that the conformation of the hydrophobic residues in the
interface controls the population of the extended and compact states.
Our structural ensembles demonstrate that pCDC25c and FFpSPR

cause distinct conformational changes in Pin1 primarily through previously proposed Path 1 and Path 2, respectively. While the MD study only examined FFpSPR and the *cis/trans*-locked isosteres, we report that changes due to pCDC25c binding also fall within these clusters, which suggests that these allosteric paths are inherent to Pin1.

Although we see conformational changes that appear to propagate from the interdomain interface to the PPIase catalytic site, we are unable to ultimately conclude that the changes are due to the contact (or lack-there-of) with the WW domain as the PPIase itself binds/isomerizes pCDC25c as well. We note, however that the binding affinity is at least one order of magnitude lower than for the WW domain and the CSPs are also relatively weak in the active site upon ligand binding[20]. Until the structural ensemble of the isolated PPIase is solved with pCDC25c bound (no solution structure of the isolated PPIase bound to any ligand has been deposited in the PDB to date), we can only hypothesize which perturbations link the WW domain to the PPIase catalytic site.

The conformational flexibility described here likely promotes the substrate interaction promiscuity in Pin1. Pin1 is known to have a large interactome; within cell cycle progression alone, Pin1 is known to interact with CDK1, CDC25c, Wee1, CENP-F, p53, and p27[21–27]. Some of the substrates interact with Pin1 only at one site, including β-catenin and NFkB[28,29]. For such cases, a role of the coexistence of compact and extended states that immediately emerges from this study is a continuous rather than a binary control of the enzymatic activity of Pin1 through ligand binding to the WW domain. Pin1 typically targets intrinsically disordered regions of substrate proteins, frequently involving multiple pS/T-P sites that vary in distance from 5 to 30 + residues[30]. Some examples of multivalent Pin1 ligands include tau, full-length CDC25c, RNA Pol II, and IRAK1[8,31]. Here, an additional role of the coexistence of these states may be a specific accommodation of ligands with multiple binding sites. The function may be directly dependent on the separation of two such binding sites, which then fine-tunes the interaction with Pin1. Alternatively, this coexistence of states may provide a mechanism to bind two different molecules, each to one Pin1 domain, which in turns repositions the two bound molecules by a shift from extended to compact state of Pin1. In these scenarios, interdomain flexibility allows Pin1 to interact with a higher variety of multivalent ligands as proposed through a "fly-casting" model[14,32]. We expect the detailed structures of ligand-bound Pin1 to be utilized to understand what peptide sequence determinants lead to differences in the interdomain equilibrium. Recent work has described the residues preceding pS/pT as most critical, with polar residues (as in peptide pCDC25c) stabilizing the extended state more than hydrophobic residues (FFpSPR)[19]. Further exploration of substrate sequence determinants and the Pin1 in-cell interactome will aid in determining the functional consequences of Pin1's malleable interdomain equilibrium.

## Methods

### Materials
The protein expression and purification, the measurements of NOE buildups, scalar couplings, structure calculations, relaxation rates, PRE, and DEER experiments were extensively described in our recent publication on apo Pin1[9].

Ligands FFpSPR and pCDC25c (sequence EQPLpTPVTDL) were synthesized by the Peptide and Protein Chemistry core at Univ. of Colorado Anschutz using Fmoc solid-phase synthesis. The N-termini were acetylated while the C-termini were amidylated to protect the peptides against degradation. HPLC was performed to purify the peptides, and samples were analyzed by LC/MS which confirmed purity >96%. The samples were lyophilized and resuspended in NMR buffer (20 mM sodium phosphate, 50 mM sodium chloride, 5 mM dithiothreitol, 0.03% sodium azide at pH 6.5) dependent on their solubility. Ligand pCDC25c was more soluble (40 mM stock) than

FFpSPR (4 mM stock) in this buffer, likely due to the hydrophobic phenylalanines in the latter. No DMSO was used.

### NMR spectroscopy
For assignment, scalar coupling, relaxation, and NOESY experiments, all samples contained $^{15}$N,$^{13}$C-labeled Pin1 in NMR buffer at pH 6.5 with 3% D$_2$O (low D$_2$O to reduce H-D exchange). While the apo sample contained ~2 mM Pin1, the amount of protein had to be reduced for the ligand-bound samples due to low concentration of FFpSPR, and to reduce aggregation with pCDC25c. Therefore, the FFpSPR-saturated sample contained 600 μM Pin1 and 3.6 mM FFpSPR and the pCDC25c-saturated sample contained 750 μM Pin1 and 4 mM pCDC25c. Apo Pin1 was assigned as described previously[9]. Using $^{15}$N-HSQC, $^{13}$C-resolved aliphatic and aromatic CT-HSQC, and a $^{15}$N/$^{13}$C-resolved [$^1$H-$^1$H] NOESY experiment, we assigned the chemical shifts of the FFpSPR- and pCDC25c-bound Pin1 based on the apo Pin1 assignment[18]. While chemical shift changes occur, the NOESY towers remain relatively intact and characteristic of each atom. The chemical shifts of FFpSPR-bound and pCDC25c-bound Pin1 have been deposited in the BMRB under accession codes 51043 and 51034, respectively. NOE buildup series were run as previously described[1] with mixing times of 24, 32, 40, 48, and 56 ms. Scalar coupling ($^3J_{HN-H\alpha}$, $^3J_{H\alpha-H\beta}$[33], and $^3J_{N-C\gamma}$) and relaxation ($R_1$ and $R_{1\rho}$ for determination of tumbling times) experiments were also recorded on these ligand-bound samples, as previously described[9].

The same cysteine-mutant constructs for PRE and DEER that were developed to measure the interdomain orientation of apo Pin1[9] were also used for ligand-bound measurements. The samples were expressed, purified, and spin-labeled as previously published[9]. For PRE, the ligands were added to be 8x the concentration of Pin1. $R_2$ PRE rates were measured on the $^{15}$N-labeled MTSL-conjugated (paramagnetic) and quenched (diamagnetic) samples M15C, N90C, S98C, and Q131C that all also contained C57A and C113D mutations. The relaxation enhancement due to the paramagnetic labels ($R_2^{sp}$) were obtained from the $R_2$ difference between the paramagnetic and diamagnetic samples[34]. The DEER samples M15C-N90C, M15C-S98C, M15C-Q131C, and N90C-Q131C were all measured with 8x ligand. All PRE and DEER constructs maintained catalytic activity, as previously published[9].

Data fitting and analysis for the ligand-bound samples were performed as previously published for apo Pin1, using TopSpin (Bruker) version 7 and VNMRJ version 4.2 Revision A (Agilent), CCPnmr version 2.4.2 (CCP), and NMRpipe Version 10.9 (NIST IBBR).[9]

### Pulsed EPR spectroscopy: Double Electron-Electron Resonance
Pulsed Double Electron-Electron Resonance (DEER) Q-band EPR measurements were recorded at 50 K with a Bruker Elexsys E580 spectrometer equipped with a home-built resonator[35] and an incoherent arbitrary waveform generator pulse channel using Xepr. A total of 40 μL of 30–80 μM MTSL-labeled Pin1 in buffer:glycerol-d$_8$ 1:1 v:v were filled in 3 mm o.d. quartz capillaries, flash frozen and stored in liquid nitrogen between measurements. Echo-detected fieldsweeps were recorded using π/2−τ−π−τ with pulse lengths $t_{\pi/2} = t_\pi/2 = 12$ ns and an interpulse delay of τ = 400 ns. DEER-EPR data were acquired either with the 4-pulse DEER (4pDEER) sequence π/2$_{obs}$−τ$_1$−π$_{obs}$−(τ$_1$+t)−π$_{pump}$−(τ$_2$−t)−π$_{obs}$−τ$_2$[36] or the 5-pulse DEER (5pDEER) sequence π/2$_{obs}$−(τ/2−t$_0$)−π$_{pump}$−t$_0$−π$_{obs}$−t−π$_{pump}$−(τ−t+δ)−π$_{obs}$−(τ/2+δ)[37] featuring a time shift δ = 120 ns to separate the refocused from the stimulated echo[38]. 4pDEER measurements were performed with monochromatic, rectangular pulses of length $t_{\pi,pump} = 12$ ns, applied at the maximum of the nitroxide Q-band spectrum and observer pulses with $t_{\pi/2,obs} = t_{\pi,obs} = 16$ ns, offset 100 MHz from the pump frequency. 5pDEER measurements were recorded with HS{1,6} pump pulses of 150 MHz width and monochromatic, rectangular observer pulses with $t_{\pi/2,obs} = t_{\pi,obs} = 32$ ns, placed 70 MHz away from the pump position. In each case, nuclear

modulations were averaged using an eight-step phase cycle with 16 ns steps.

All DEER data were analyzed using an earlier version of DeerLab (DL) based on Matlab[39] (release 0.9.2, available under https://github.com/JeschkeLab/DeerLab-Matlab) by modelling the background decay by a stretched exponential function (bg_strexp, $B(t) = \exp(-\kappa|t|^d)$) with the decay rate $\kappa$ and the stretch factor $d$ restraint to 0.02–1 $\mu s^{-1}$ and 0.9–1.2, respectively. 4pDEER and artefact corrected 5pDEER data[40] were analyzed using the single-pathway 4pDEER experiment model (ex_4pdeer) including the modulation depth $\lambda$ as a fit parameter. For the analysis of the primary 5pDEER data, the multi-pathway 5pDEER model (ex_5pdeer) was applied. This model consists of an unmodulated, the 5pDEER pathway and 4pDEER pathway (artefact that refocused at time $T_0^{(2)}$) with amplitudes $\Lambda_0$, $\lambda_1$ and $\lambda_2$, respectively. Parameter-free and Gaussian distance distributions $P(r)$ were computed via Tikhonov regularization, using generalized cross-validation (GCV) to select the optimal regularization parameter. Bootstrapping with 200 samples produced converged 95% confidence intervals (CI), listed in Supplementary Tables 2–4 for all fitting parameters. Neural network analysis was performed by DEERNet SVN rev 5662, using the Comparative DEER Analyzer feature of DeerAnalysis 2021.

### Multi-state structure calculations

As with apo Pin1 (PDB ID: 7SA5), we combined all PRE, eNOE, and scalar coupling restraints and calculated multi-state ensembles of ligand-bound Pin1. To account for the time- and ensemble-averaged nature of the probes, these restraints must be fulfilled by the average of the back-calculated contributions from each individual state. To allow large spatial sampling between the two domains, we (1) calculated the structure of the WW domain and then (2) froze the WW angles and used them as an input to determine the PPIase structure and interdomain positions[9]. In contrast to our original protocol, we omit RDC restraints from our ligand-bound ensembles as we saw that RDCs had a negligible effect on our resulting apo Pin1 ensemble. This has the advantage that we can exclude potential interactions between ligands and alignment media, or that the relative domain orientations are impacted by the induced alignment. The PDB IDs for FFpSPR- and pCDC25c-bound two-state ensembles are 7SUQ and 7SUR, respectively.

Similar to our multi-state ensemble calculations of apo Pin1, the CYANA target function (TF, proportional to the sum of squared violations) was high for the single state structures (~300 Å² for all three ensembles) but decreased significantly upon addition of a second state to fulfill all the structure restraints (Supplementary Table 5). Note that the single-state structure is more akin to an averaged structure. All three Pin1 single-state structures resulted in a compact conformation as all the interdomain NOEs needed to be fulfilled. When a second state was allowed, an extended and a compact state were generated with all the interdomain NOEs fulfilled in the compact state.

While the interdomain NOEs and PREs were sufficient in orienting the two domains in apo Pin1 to agree with the DEER distribution, the position of the extended states did not fulfill longer distances from the DEER distance distributions with the ligand-bound ensembles. Therefore, we calculated population- and fluctuation-averaged distances between the Cβs of the spin-labeled residues based on the DEER distance distributions and used them as restraints in our structure calculations. A tolerance of ±5 Å was added to these effective distances to use as upper and lower limit restraints. While we optimized and lowered the PRE distance restraint weight to 0.01 (relative to the NOE weight of 1)[9], we kept a weight of 1 for these DEER effective distance restraints as there were only 3 distances added for the calculation, and these restraints did not appreciably increase the target function. Importantly, the addition of the DEER effective distances adjusted the interdomain positions but did not change the intradomain structural correlations. When we calculated a ligand-bound two-state ensemble with the interdomain NOE, PRE, and averaged DEER restraints (see Supplementary Information), we obtained a satisfactory match to the experimental DEER distributions, with only a slight under-representation of small populations of longer distances.

### Reporting summary

Further information on research design is available in the Nature Research Reporting Summary linked to this article.

## Data availability

Source data are provided with this paper. The two-state structural ensembles of FFpSPR- and pCDC25c-bound Pin1 generated in this study have been deposited in the Protein Data Bank under the PDB IDs 7SUQ, 7SA5 and 7SUR. The chemical shifts of FFpSPR- and pCDC25c-bound Pin1 collected in this study have been deposited in the Biological Magnetic Resonance Data Bank under the BMRB accession codes 51043 and 51034, respectively. The entire CYANA structure calculations, including intermediate structures, are available in the Dryad repository, with the identifier [https://doi.org/10.5061/dryad.bvq83bkcf]. All other datasets generated during and/or analyzed during the current study are available from the corresponding author on reasonable request.

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

## Acknowledgements

We thank Dr. David Jones (University of Colorado) for support in NMR spectroscopy. This project was supported by NIH Grant R01 GM130694, a start-up package by the University of Colorado to B.V., University of Colorado Cancer Center Support Grant P30 CA046934, NIH Biomedical Research Support Shared Grant S10 OD025020, NIH Grant R21 AI171827 to M.H., a Günthard foundation PhD scholarship to J.S., and SNF Grant 200020_169057 to G.J.

## Author contributions

A.B., J.S., and F.B. performed all experiments and data analysis with the help of M.A.H. (NMR) and Y.P. (EPR), and A.B., J.S., M.A.H., F.B., Y.P., G.J., and B.V. interpreted the data. A.B., G.J. and B.V. conceptualized the study. A.B., J.S. and B.V. wrote and revised the manuscript. All authors provided comments on the manuscript.

## Competing interests

The authors declare no competing interests.

## Additional information

**Peer review information** *Nature Communications* thanks Xuanjun Ai and other anonymous reviewer(s) to the peer review of this work. Peer review reports are available.

