## [Peer Review File · Nature Communications]

REVIEWER COMMENTS

Reviewer #1 (Remarks to the Author):

The authors describe two-state structures of Pin1 bound to two different peptides, one synthetic and the other from a natural protein. These structures are derived from an impressive set of experimental data including eNOEs, scalar couplings, PREs, and DEER data. Furthermore, they show that binding to one of the peptides, pCDC25c, destabilizes a compact form of the protein. Even though there is some uncertainty in the populations, this data is convincing (other than my concern with the interpretation of the slow exchange peak populations noted below). The paper suggests some intriguing mechanisms for interdomain allostery whose potential significance would be better revealed by additional data analysis and better presentation.

Major Comments:

The authors have not sufficiently eliminated the possibility that the specific structural correlations in the two-state ensembles are a result of modeling artifacts. For instance, in the 2-state ensembles, are there eNOEs for specific atom pairs (for example between L141 and some other residue) that support the correlation (or lack thereof) observed in the models? Showing data for such an eNOE might also shed light on this observation: "However, in the FFpSPR-bound form (Figure 5a), this difference between the two states is only maintained in the WW domain's interface (residues 28-30)"

What effect do the RDCs have on restraining apo Pin1? Would the differences between the apo and ligand bound forms shown in Figures 4-6 still be as pronounced if the RDC data were excluded and the exact same types of data were used to refine both apo and ligand bound Pin1? Perhaps the lack of RDC data for the ligand bound forms causes the majority of the differences and not the presence of the ligand. (I do note that Figure 6A shows up to 10 Å RMSD between the two ligand bound forms, both of which exclude RDC data.)

Going further, in the ideal case, the authors would show ensembles refined with identical subsets of the restraints (i.e. with the same set of eNOE atoms pairs, J-couplings, and PRE restraints). If the conclusions from the paper depend on restraints that are only present in a one or two of the structures, then some verification would be warranted that the absence of those restraints in the other structure(s) are indeed due to them truly being absent and not because they were missed in the manual/automatic assignment process.

It was very difficult to compare what is described in the text to what is shown in Figures 3-6. This difficulty reaches a head with Figure 6E, where three different orientations of the PPIase are shown without a very clear way of understanding the relationship between them. The authors seem intent on showing/describing almost every single difference between the structures rather than focusing on the bigger picture conclusions. This reader got lost in the trees and had a lot of difficulty seeing/understanding the forest. Without a lot of detailed study (or prior familiarity with Pin1), the general readership of Nature Communications may find it difficult to see how the structural data justifies the conclusions. It may be helpful to focus on a small set of orientations that are first introduced to the reader with an overview figure showing the overall protein context (with hypothetical peptide coordinates modeled in, active site, etc.), that are then used and labeled consistently in subsequent figures.

I can't easily see how the author's data supports or contrasts the previously described "path 1" or "path 2". The paths are only ever shown in Figures S6. The plethora of different orientations used in the paper make it very difficult to somehow compare with Figures 3-6.

The slow exchange peak analysis (Figure S5) seems to only show a ligand-driven effect for the mutants and not the WT protein. If this is reflective of the compact/extended states, it would seem to suggest that the mutations/spin labels significantly affect the protein energy landscape depending on where the spin labels are placed, which casts doubt on the rest of the structural analysis.

Minor Comments:

The two different overall conformations of Pin1 are never shown in the paper (with the exception of Figure 5B), nor are the probable coordinates of the peptide ligands, which could be superimposed from the cited cocrystal structure. This would be very helpful for seeing the big picture.

What kind(s) of RMSD is(are) shown in each of the figures (backbone, including side chains, etc.)?

There is no Figure 5E as stated here: "Within "path 2", we see a major change in the orientation of the $\alpha 1$ helix and the subsequent rearrangement of the PPIase catalytic loop (residues 65-80) upon addition of FFpSPR (Figures 5e and 6a)."

When residue numbers are mentioned in the text and a figure panel is referred to, I should be able to easily find those residues on that particular panel. While that is possible in some cases, in quite a few cases the residue numbers mentioned in the text are not visible in the figures, making understanding the paper more difficult.

Reviewer #2 (Remarks to the Author):

The manuscript by Born et al. combines a very powerful set of tools, including NMR NOE constraints, relaxation enhancement, and DEER spectroscopies to probe the intra- and inter-domain structural changes in Pin1, a peptide-proline isomerase involved in cell regulation. The study appears to offer much new information on the effect of two classes of peptide ligands on the inter-domain association. The methods are powerful and the system is of importance. However, the paper suffers greatly from being overly dense, disorganized and nearly impossible to evaluate. The main problem is the combination of results with discussion which muddies the distinction between what has been demonstrably observed and how it can be interpreted. The figures are far too complex to follow and present a large ensemble of conformational sub states that don't provide a clear basis for drawing the conclusions made about the distinct sub states for the compact and extended conformations that are discussed. It would be more important to present the observations and what they can be shown to exist in these two conformations instead of a large and continuous ensemble of states. Other areas where this lack of distinction between observation and interpretation are:

1. line 169 - 170 where changes in the quantitative ratio of two conformations are taken from DEER analysis. How does DEER analysis provide quantitative analysis of populations?
2. Figures 3 and 4 appear to show the same things with just different color schemes and views.
3. Figure 4b legend, "RMSD between the mean pCDC25c bound and app structures is plotted..." Is the for all the members of the ensemble or separated into the compact and extended forms?
4. Its not clear from this and other figures what provides the distinction between the compact and extended conformations. It is possible to see changes in structure between apo and ligand bound states, but it is not clear to see from the figures that the compact and extended conformations both exist for each of the apo and ligand bound forms. That is, there appears to be no distinct overlap of similar states for both forms. This is where it would be best to provide an analysis of the ensemble of structures to show that statistically significant subpopulations exist and that they overlap for the app and ligand bound states, before going ahead and discussing the implications.

These powerful methods by well qualified researchers are clearly discovering important changes in the protein structures. However, a more clear presentation is needed of what conformations are really distinct and how much they share with each other in the different states. Without this, it is difficult to evaluate the significance of the conclusions.

Reviewer #3 (Remarks to the Author):

The authors applied a method they developed before to get the structural ensembles of the compact and extended states of Pin1 with two different ligands. A detailed structural analysis revealed that different ligand bindings correlate to different interdomain allostery. This study is very novel as it verified the possible allosteric mechanism of ligand regulation by the important two-domain enzyme Pin1, which may attract much attention from readers in different research fields.

Suggestions: a graph of Pin1 with secondary structures labeled should be provided, so that the readers could easily follow. In addition, before or after Fig. 6, a scheme should be provided to explain how different interdomain allostery happens when Pin1 binds to pCDC25c or FFpSPR ("path 1" and "path 2").

My concerns are as follow:

1. The constructs 15-90, 15-98 and 15-131 all use the MTSL label in residue 15. However, 15 is too close to residues R14 and R17 involving in ligand binding. More evidences are needed to exclude its direct influence on ligand binding and its indirect influence on PPIase's catalysis, such as K_d and KEXSY.
2. Is the free energy of the extended state high than the compact state as authors state that pCDC25c but not FFpSPR stabilizes the extended state? Which state contributes the PPIase catalytic capacity, the extended state or the compact state?
3. Errors of π in Fig.2b.
4. Because rotamers of key residues are used to deduce different allosteric mechanism, could the authors explain how they get the precise conformation of residues, such as A140 and L141 (lines 197, 210, 264, ...), on interdomain interface? In another word, how confident for those described "point away", "point down" or "point back"? Are there NOE evidences between A140/L141 and residues on interface of WW, in apo and FFpSPR-bound Pin1?
5. The ligand binding loop "folds" upon binding pCDC25c, in line 274, page 8. If this hypothesis is right, could the authors provide experimental evidences to support it, such as, intermolecular NOEs and CSP? In addition, how about NOEs and CSP for Pin1 and FFpSPR?
6. line 375-382, page 12. Could the authors provide NOEs and information of backbone dynamics for the residues in the catalytic loop of Pin1 in the presence of pCDC25c and FFpSPR, so as to verify the rigidity of the catalytic loop in the two complexes?

7. Could the authors design experiments to verify the “path 1” hypothesis after some residues in alfa-1 helix for FFpSPR binding are mutated?

8. A paragraph should be added to discuss why isomerase Pin1 has the flexibility in dealing with different ligands.

REVIEWER COMMENTS

Reviewer #1 (Remarks to the Author):

The authors describe two-state structures of Pin1 bound to two different peptides, one synthetic and the other from a natural protein. These structures are derived from an impressive set of experimental data including eNOEs, scalar couplings, PREs, and DEER data. Furthermore, they show that binding to one of the peptides, pCDC25c, destabilizes a compact form of the protein. Even though there is some uncertainty in the populations, this data is convincing (other than my concern with the interpretation of the slow exchange peak populations noted below). The paper suggests some intriguing mechanisms for interdomain allostery whose potential significance would be better revealed by additional data analysis and better presentation.

Major Comments:

The authors have not sufficiently eliminated the possibility that the specific structural correlations in the two-state ensembles are a result of modeling artifacts. For instance, in the 2-state ensembles, are there eNOEs for specific atom pairs (for example between L141 and some other residue) that support the correlation (or lack thereof) observed in the models? Showing data for such an eNOE might also shed light on this observation: "However, in the FFpSPR-bound form (Figure 5a), this difference between the two states is only maintained in the WW domain's interface (residues 28-30)"

See below.

What effect do the RDCs have on restraining apo Pin1? Would the differences between the apo and ligand bound forms shown in Figures 4-6 still be as pronounced if the RDC data were excluded and the exact same types of data were used to refine both apo and ligand bound Pin1? Perhaps the lack of RDC data for the ligand bound forms causes the majority of the differences and not the presence of the ligand. (I do note that Figure 6A shows up to 10 Å RMSD between the two ligand bound forms, both of which exclude RDC data.)

See below.

Going further, in the ideal case, the authors would show ensembles refined with identical subsets of the restraints (i.e. with the same set of eNOE atoms pairs, J-couplings, and PRE restraints). If the conclusions from the paper depend on restraints that are only present in a one or two of the structures, then some verification would be warranted that the absence of those restraints in the other structure(s) are indeed due to them truly being absent and not because they were missed in the manual/automatic assignment process.

The first three points addressed by the reviewer are closely related, and we are going to address them as one in the following. The underlying question of all three is whether the information we make use of to calculate our structural ensembles is really in the data rather than caused by structure calculation artifacts.

We have conducted extended analysis to address this question:

A near-identical restraints data set for different forms of Pin1 would be expected for very similar structures. Conversely, different conformations result in different data sets for two reasons: First, as opposed to torsion angle-dependent restraints, distance-dependent restraints are expected to be lost for some conformations because their cross peaks maybe lost in noise (NOEs) or their quenching is modified in different ways (PREs), while others may appear. Second, differences in structures result in slightly different chemical shifts such that some peaks become overlapped, while others will be resolved. Thus, even torsion angular-dependent restraints will form non-identical restraint data sets. We determined the percentages of the restraints used for the calculations complexes with FFpSPR and pCDC25c that were shared with the restraints used for the other complex (*Table 1*). Indeed, we observe many differences in the data sets (even though we used near-identical conditions for all samples).

Table 1. Comparing the complexes with FFpSPR and pCDC25c, the following percentages of the restraints are shared with the other complex:

	FFpSPR files	pCDC25c files
NOE upl,lol	63	55
PRE upl,lol	76	80
EPR upl,lol	100	100
J coupling, cco	73	87

angle from CS, aco	100	100
-----	-----

Comparisons between each complex and apo form yield similar tables. The relatively small overlap between data sets already demonstrates clear differences between the different structures. Identical structures of the real molecules (that should not result in differences in structure calculation) would give a much larger overlap.

Now addressing point 3 raised by the reviewer, we ran structure calculations with only overlapping data sets.

Given the severely reduced numbers of restraint of such data sets, it is somewhat surprising that we still observe the differences between the various ensembles, although somewhat attenuated. Perhaps an extreme example where we expected the differences only to be caused by restraints that are NOT shared, is the ligand-specific bend of the ligand binding loop of the WW domain. However, although there are many NOEs unique to only the structure in complex with one ligand, we clearly see the bend when using only shared restraints:

Table 2. eNOEs that drive bending the loop in complex with pCDC25c and are not shared with FFpSPR, identified by systematically omitting in structure calculations:

14 ARG HA	15 MET H	2.22
17 ARG HA	20 GLY H	3.42
17 ARG HA	21 ARG H	3.39
18 SER H	20 GLY H	3.37
18 SER HA	20 GLY H	5.96
19 SER HA	22 VAL QG2	5.13
22 VAL HB	23 TYR H	3.81
22 VAL QG1	24 TYR HA	4.26
22 VAL QG2	24 TYR QD	7.13
23 TYR H	24 TYR QE	7.13

Figure 1. Superposition of WW domains of the two-state structural ensembles of Pin1 complexes with pCDC25c (blue) and FFpSPR (red) calculated only from shared eNOE data sets.

We note that the bending of the WW domain has been independently observed by Peng et al. through the use of PREs induced by a label that is ideally positioned to achieve this task (Zhang et al., "Coupled intra- and interdomain dynamics support domain cross-talk in Pin1", 2020, J. Biol. Chem., 295, 16585). Another important aspect is the tilt of the alpha helix 1 being part of allostery pathway 2 in the PPIase domain. Again, we still see the difference, albeit weakly, when only using shared data sets:

Figure 2. Superposition of alpha helix 1 of the two-state structural ensembles of Pin1 complexes with pCDC25c (blue) and FFpSPR (red) calculated only from shared eNOE data sets.

In general, the structural differences are caused by a combination of differences in restraints that are shared AND restraints that are not shared.

Another consistency test is to check the effect that the RDCs have on the apo structure (This addresses point 2 above raised by the reviewer). In fact, we have already addressed that question in our previous JACS paper (Born et al., “Reconstruction of Coupled Intra- and Interdomain Protein Motion from Nuclear and Electron Magnetic Resonance”, 2021, JACS, 143, 16055), where we established the method and calculated multi-state ensembles for apo Pin1. In an analysis of the role of individual restraints, we systematically omitted subsets of restraints. We found that, as opposed to omitting PREs or interdomain NOEs, omission of the RDCs had a negligible effect on the resulting structural ensembles. This is the reason why we did not use RDCs for the calculation of the ligand-bound ensemble, which has the advantage that we don’t have to worry about potential interactions between ligands and alignment media, or that the relative domain orientations are impacted by the alignment (now mentioned in Method sections). Importantly, as can be seen in *Figure 3*, the difference

between compact and extended states in the interdomain interface is maintained.

Figure 3. Two-state structural ensembles of apo Pin1 calculated without RDCs.

Importantly, and as the reviewer pointed out, both complex structures were calculated without RDCs, and show differences between each other to a similar extent as to the free form, again confirming that relevant differences are not caused by RDCs.

Although we originally intended to present the results rather than technical aspects of the structure calculations, we agree that it is important to address some of these aspects. Therefore, we took advantage of the shortening of the structural discussion (see below) and added a new paragraph that summarizes the points discussed above to the main text:

"NMR restraints indicate global structural changes upon ligand binding

In order to deduce the inter- and intradomain conformational changes induced by ligand binding, we calculated the two-state NMR ensemble using eNOEs, scalar couplings, PRE and DEER restraints as previously described¹⁸. As for apo Pin1¹⁸, a two-state ensemble resulting in compact and extended states is

necessary for both ligand-bound complexes to satisfy all the experimental data (see Methods for details on the structure calculation and Supplemental Table 5 for structural statistics). A simple analysis of the restraints foreshadows the global structural changes that we will obtain for the two-state ensembles (Figure 3a). Comparing the restraint input sets of FFpSPR-, pCDC25c-bound and apo Pin1 to the other two sets shows that 52-63% eNOE, 72-84% PREs, and 60-91% scalar couplings are shared. In part, loss of shared restraints is caused by changes in chemical shifts due to binding (see Supplemental Figure 1c) that may resolve some overlapped peaks and overlap some resolved peaks. In addition, many NOE cross peaks appear or disappear due to changes in distances, which results in the lowest shared portion among the different restraint types. Supplemental Figure 4 shows examples of comparisons between NOESY spectra from FFpSPR-, pCDC25c-bound and apo Pin1, and eNOE restraints in the WW domain that are unique to one structure with respect to any of the other two. To determine whether the specific conformational changes in our structural ensembles are only a result of these unique restraints, we also calculated ensembles with identical subsets of restraints. Even with omission of unique restraints, the ligand-dependent conformational changes are still present, but generally with a reduced difference between the states. This indicates that the structural changes are encoded both in shared *and* unique restraints.”

To appreciate the nature of restraints that are not shared between the different ensembles, we show examples of specific NOEs for all three structures that directly demonstrate the differences in Supporting Figure S4 (This addresses point 1 above raised by the reviewer, we show among other examples interdomain eNOEs connecting regions 28-30 and 140-141, as specifically mentioned by the reviewer).

Figure 4 (is Figure S4 in revised manuscript). eNOEs unique to only apo or one of the complexes plotted on the WW domain to visualize the information content of the eNOE sets (The NOEs were compared between FFpSPR-bound and pCDC25c-bound (red/blue distance plot), between pCDC25c-bound and apo (blue/yellow), and between FFpSPR-bound and apo measurements (red/yellow), with the respective unique NOEs plotted on the WW domain), and direct comparisons of NOESY towers with important differences in cross peaks.

It was very difficult to compare what is described in the text to what is shown in Figures 3-6. This difficulty reaches a head with Figure 6E, where three different orientations of the PPIase are shown without a very clear way of understanding the relationship between them. The authors seem intent on showing/describing almost every single difference between the structures rather than focusing on the bigger picture conclusions. This reader got lost in the trees and had a lot of difficulty seeing/understanding the forest. Without a lot of detailed study (or prior familiarity with Pin1), the general readership of Nature Communications may find it difficult to see how the structural data justifies the conclusions. It may be helpful to focus on a small set of orientations that are first introduced to the reader with an overview figure showing the overall protein context (with

hypothetical peptide coordinates modeled in, active site, etc.), that are then used and labeled consistently in subsequent figures.

We agree with the reviewer that our comparisons were very dense and hard to follow. Therefore, we came up with a re-arranged and stripped-down version in the main text, focusing on the core aspects. For the interested reader, we relegated details to the Supporting Information.

In detail, we reduced Figures 3-6, all of which present structures, to two figures (3 and 4 in the revised version) with less details. This goes along with a shorter discussion that only focuses on the core findings.

We also made more use of simple schematic renderings: First, we supplemented the introduction figure (Figure 1 in the revised version) with simple presentations of previously solved X-ray and NMR structures to indicate the ligand binding sites and give a simple visualization of complex and extended states:

Figure 5 (is Figure 1 in revised manuscript). In a and b, we now show a previously solved compact and complexed X-ray structure, and our previously solved eNOE apo Pin1 structure (two conformers each consisting of two states).

Second, we added a new figure (Figure 5 in the revised version) that summarizes our findings in a simple cartoon:

Figure 6 (is Figure 5 in revised manuscript). Proposed model of ligand-dependent conformational changes.

I can't easily see how the author's data supports or contrasts the previously described "path 1" or "path 2". The paths are only ever shown in Figures S6. The plethora of different orientations used in the paper make it very difficult to somehow compare with Figures 3-6.

We indicate the pathways clearer in the revised manuscript. See *Figure 6* (which is *Figure 5* in revised manuscript) and also new *Figure 3b* in the revised manuscript. We also describe Paths 1 and 2 as proposed by the previous MD study in more detail in the the introduction.

The slow exchange peak analysis (*Figure S5*) seems to only show a ligand-driven effect for the mutants and not the WT protein. If this is reflective of the compact/extended states, it would seem to suggest that the mutations/spin labels significantly affect the protein energy landscape depending on where the spin labels are placed, which casts doubt on the rest of the structural analysis.

To avoid a very technical explanation, we oversimplified the discussion of the effects of addition of ligand on exchange peaks in the original version. As a matter of fact, there are typically multiple exchange peaks (for example, see *Figure 3* in the Supplemental Information). We measured the peak heights of the most separate exchange peaks to generate the plot in c). We are not able to conclusively assign specific conformations to specific exchange peaks.

Additionally, the relevant timescales may also cover fast exchange that would not result in separate exchange peaks. A paper that got published during our revision focuses on chemical shift changes upon addition of various ligands to Pin1 (Chen, "A specific Thr-Pro motif generates interdomain communication bifurcations of two modes of Pin1 in solution nuclear magnetic resonance", 2022, *Biochemistry*, in press). The author encountered the same difficulty with assigning extended/compact populations to exchange peaks, literally stating that "...the slow exchange NMR peaks do not give us a clear indication of the specific behavior of the major and minor peaks corresponding to compactness and dispersion." Therefore, we do only loosely rely on the analysis and rely on the much more reliable EPR analysis and inter domain NOEs.

In the revised version, we point out the difficulty with exchange peak analysis clearer and refer to the analysis as "qualitative":

"Independent qualitative assessment of these population shifts comes from slow-exchange NMR peaks observed for residues located in the interdomain interface (Supplementary Figure 3e-g)^{3,18,19}. As multiple exchange peaks exist, we are unable to conclusively assign specific conformations to specific exchange peaks. Nevertheless, the populations of the compact and extended states from the 15-90 and 15-98 DEER distance distributions are similar to the major and most separate minor slow-exchange peak populations from the N90C (~0.66:0.34) and S98C PRE mutant constructs (~0.78:0.22). Thus, we suggest that these major and minor peaks emanate predominantly from the compact and extended conformations, respectively. This is further supported by the absence of interdomain NOESY peaks for the minor exchange peaks, as expected for extended states. Because the extended state does not contribute appreciably to the interdomain NOEs, we cannot discount the possibility that though the major peaks stems primarily from the compact state also the extended state contributes. Furthermore, we also cannot discount that the relevant timescales for interdomain mobility also cover fast exchange which does not result in separate exchange peaks. Therefore, the most separate slow-exchange minor peaks set a lower limit to the extended population. ."

Minor Comments:

The two different overall conformations of Pin1 are never shown in the paper (with the exception of Figure 5B), nor are the probable coordinates of the peptide ligands, which could be superimposed from the cited cocrystal structure. This would be very helpful for seeing the big picture.

We now show the previously published overall apo Pin structure in the introduction figure (Figure 1 in the revised manuscript). In the new Supporting Figure S5, where we compare the structure of the two complexes, we also show overall structures.

What kind(s) of RMSD is(are) shown in each of the figures (backbone, including side chains, etc.)?

The reviewer is talking about the RMSD plots in Fig 3 a and Fig 4a, which are of backbone and side-chain. It is now specific in the text that the RMSD includes side-chain.

There is no Figure 5E as stated here: "Within "path 2", we see a major change in the orientation of the α_1 helix and the subsequent rearrangement of the PPIase catalytic loop (residues 65-80) upon addition of FFpSPR (Figures 5e and 6a)."

We added this to the new Figure 3b.

When residue numbers are mentioned in the text and a figure panel is referred to, I should be able to easily find those residues on that particular panel. While that is possible in some cases, in quite a few cases the residue numbers mentioned in the text are not visible in the figures, making understanding the paper more difficult.

In our simplified figures, we clearly label these residues.

Reviewer #2 (Remarks to the Author):

The manuscript by Born et al. combines a very powerful set of tools, including NMR NOE constraints, relaxation enhancement, and DEER spectroscopies to probe the intra- and inter-domain structural changes in Pin1, a peptide-proline isomerase involved in cell regulation. The study appears to offer much new information on the effect of two classes of peptide ligands on the inter-domain association. The methods are powerful and the system is of importance. However, the paper suffers greatly from being overly dense, disorganized and

nearly impossible to evaluate. The main problem is the combination of results with discussion which muddies the distinction between what has been demonstrably observed and how it can be interpreted. The figures are far too complex to follow and present a large ensemble of conformational sub states that don't provide a clear basis for drawing the conclusions made about the distinct sub states for the compact and extended conformations that are discussed. It

would be more important to present the observations and what they can be shown to exist in these two conformations instead of a large and continuous ensemble of states. Other areas where this lack of distinction between observation and interpretation are:

We agree with the reviewer that the presentation was overly dense and went too much into details. This concern is the same as raised by reviewer 1. As explained above, we have substantially simplified the text, and reduced in numbers and simplified the figures presenting the structures.

1. line 169 - 170 where changes in the quantitative ratio of two conformations are taken from DEER analysis. How does DEER analysis provide quantitative analysis of populations?

DEER data provides population probability density as a function of the distance between the two electron labels. When we observe two main clusters we identify those with the compact and extended states. The integral of probability density over these clusters (or the area of Gaussian distribution fits) over these clusters then provide populations for the compact and extended states.

2. Figures 3 and 4 appear to show the same things with just different color schemes and views.

As part of the simplification of the presentation, we have these figures no longer in the old form. Some of them are now (in new presentation) in the Supporting Information.

3. Figure 4b legend, "RMSD between the mean pCDC25c bound and app structures is plotted..." Is the for all the members of the ensemble or separated into the compact and extended forms?

We describe this clearly in the relevant figure captions of the revised version. For Figure 3a: "RMS deviation (including side-chain) between the mean FFpSPR- and pCDC25c-bound structures (all conformers) is plotted versus residue number". And for Figure 4a: "RMSD (including side-chain) between mean compact and extended states of domains is plotted versus residue number with error bars depicting standard deviation of conformers."

4. Its not clear from this and other figures what provides the distinction between the compact and extended conformations. It is possible to see changes in structure between apo and ligand bound states, but it is not clear to see from the figures that the compact and extended conformations both exist for each of the apo and ligand bound forms. That is, there appears to be no distinct overlap of similar states for both forms. This is where it would be best to provide an analysis of the ensemble of structures to show that statistically significant subpopulations exist and that they overlap for the app and ligand bound states, before going ahead and discussing the implications.

We disagree with the reviewer. Our DEER data (Figure 2) clearly shows two clusters of distances that we identify with the compact and extended states. These clusters are present in apo form and in complex with both ligands. We then compare the two states within the domains. The reviewer is correct in that we do not simply observe the exactly same two states in apo and the two bound forms. Given the nature of our two-state structures, this can't be expected as the two states are the best discrete representation of the true distribution. However, there are regions where we clearly see conservation of these two states, and others without conservation. We attempted to dissect this and present in great detail in the original version, but we realize that we had to simplify this.

These powerful methods by well qualified researchers are clearly discovering important changes in the protein structures. However, a more clear presentation is needed of what conformations are really distinct and how much they share with each other in the different states. Without this, it is difficult to evaluate the significance of the conclusions.

We thank the reviewer for his appreciation of the power of our method. For addressing the presentation, see above.

Reviewer #3 (Remarks to the Author):

The authors applied a method they developed before to get the structural ensembles of the compact and extended states of Pin1 with two different ligands. A detailed structural analysis revealed that different ligand bindings correlate to different interdomain allostery. This study is very novel as it verified the possible allosteric mechanism of ligand regulation by the important two-domain enzyme Pin1, which may attract much attention from readers in different research fields.

Suggestions: a graph of Pin1 with secondary structures labeled should be provided, so that the readers could easily follow. In addition, before or after Fig. 6, a scheme should be provided to explain how different interdomain allostery happens when Pin1 binds to pCDC25c or FFpSPR ("path 1" and "path 2").

We now provide cartoon rendering in Figure 1 to introduce the system, and in Figure 5 to summarize our findings in simple terms.

My concerns are as follow:

1. The constructs 15-90, 15-98 and 15-131 all use the MTSL label in residue 15. However, 15 is too close to residues R14 and R17 involving in ligand binding. More evidences are needed to exclude its direct influence on ligand binding and its indirect influence on PPlase's catalysis, such as K_d and K_{EXSY} .

We have measured the activity of all mutants. None of the mutants experiences major changes (the largest reduction in activity come with the Q131C mutations, but still much less than an order of magnitude). We published the data in Born et al., "Reconstruction of Coupled Intra- and Interdomain Protein Motion from Nuclear and Electron Magnetic Resonance", 2021, JACS, 143, 16055:

Table 3. Isomerization rates measured using exchange spectroscopy (EXSY)

Pin1 Variant	k_{ct} (s^{-1})	k_{tc} (s^{-1})	k_{EXSY} (s^{-1})
WT	43.77 (1.17)	4.54 (0.38)	48.31(1.55)
AD+M15C	44.11 (4.22)	4.00 (0.42)	48.11 (4.64)
AD+N90C	42.09 (6.01)	3.95 (0.36)	46.04 (6.37)
AD+S98C	23.27 (1.34)	2.94 (0.45)	26.21 (2.79)
AD+Q131C	5.25 (0.45)	1.34 (0.08)	6.59 (0.53)

AD+M15C+N90C	35.03 (2.00)	4.59 (0.20)	39.63 (2.20)
AD+M15C+S98C	42.61 (6.67)	4.18 (0.53)	46.79 (7.20)
AD+M15C+Q131C	21.04 (2.09)	1.74 (0.13)	22.78 (2.22)
AD+N90C+Q131C	10.27 (0.16)	1.44 (0.08)	11.71 (0.24)
AD+S98C+Q131C	8.71 (0.91)	0.51 (0.05)	9.21 (0.96)
AD+N90C+S98C	25.49 (13.33)	1.15 (0.58)	26.64 (13.91)

We add to "NMR spectroscopy" methods: "All PRE and DEER constructs maintained catalytic activity, as previously published⁹."

2. Is the free energy of the extended state high than the compact state as authors state that pCDC25c but not FFpSPR stabilizes the extended state? Which state contributes the PPIase catalytic capacity, the extended state or the compact state?

The extended state is associated with higher activity. In fact, cleaving off the WW domain, which can be seen as the "most extended state", results in highest activity. That being said, it appears that both states are catalytic. Given that the population of the compact state is higher in apo form, we assume that its free energy is lower than the one of the extended state.

3. Errors of Pi in Fig.2b.

Because we calculated the 95% confidence intervals, we have asymmetric errors. Indicating such errors (which would require errors for population, and distances) would require many more numbers. Therefore, we show the errors in the table in the Supporting Information. We mention this in the figure caption of Figure 2 in the revised version. The 95% confidence interval for the populations ranges between 3-7%, we added to Figure 2 caption that the error is typically +/-0.03 for all measurements.

4. Because rotamers of key residues are used to deduce different allosteric mechanism, could the authors explain how they get the precise conformation of residues, such as A140 and L141 (lines 197, 210, 264, ...), on interdomain interface? In another word, how confident for those described "point away", "point down" or "point back"? Are there NOE evidences between A140/L141 and residues on interface of WW, in apo and FFpSPR-bound Pin1?

The crucial (and non-trivial) step is the stereospecific assignment. We made a special effort to assess rotamers as well as possible. We designed and published an experiment to better obtain Ha-Hb_{1,2} scalar couplings for a protein of the size of Pin1 and combined this information with eNOEs (Born et al., "Efficient Stereospecific Hb_{2/3} NMR Assignment Strategy for Mid-Size Proteins", 2018, *Magnetochemistry* 4, 25). As such, our data is superior to standard data sets consisting mostly of conventional NOEs. We actually demonstrated that conventional NOEs are not sufficient for the purpose of obtained correct Chi angles or even rotamers (Orts et al., Stereospecific assignments in proteins using exact NOEs 2013, *J Biomol NMR*, 57, 211).

The assignments and resulting conformations are dependent on a combination of eNOEs and J couplings. Some differences can already be seen from the J couplings alone: The HN-Ha scalar couplings of residue 140 are different (beyond the uncertainty) between the different complexes, and the Ha-Hb_{1,2} scalar couplings are different between the free form and the FFpSPR- bound form:

free	ALA140 H-HA	8.57 +/-0.55
free	LEU141 H-HA	5.59 +/-0.62
free	LEU141 HA-HB ₂	3.47 +/-1.47
free	LEU141 HA-HB ₃	8.36 +/-1.15
FFpSPR	ALA140 H-HA	9.21 +/-0.75
FFpSPR	LEU141 H-HA	6.45 +/-0.74
FFpSPR	LEU141 HA-HB ₂	3.04 +/-1.62
FFpSPR	LEU141 HA-HB ₃	6.03 +/-1.16
pCDC25c	ALA140 H-HA	7.85 +/-0.77
pCDC25c	LEU141 H-HA	6.89 +/-0.79
pCDC25c	LEU141 HA-HB ₃	5.38 +/-1.07

5. The ligand binding loop "folds" upon binding pCDC25c, in line 274, page 8. If this hypothesis is right, could the authors provide experimental evidences to support it, such as, intermolecular NOEs and CSP? In addition, how about NOEs and CSP for Pin1 and FFpSPR?

We have addressed this question in our response to reviewer 1 (see above). As detailed, this bending is partially driven by un-shared eNOEs, and in part by eNOEs that are shared between both complexes.

We note that CSP analysis is not helpful because the ligand itself also induces CSPs, in addition to conformational changes of Pin1.

We also point out that the Peng lab also reported the bending, based on PRE data from a mutant that is perfectly suited for that study, and is orthogonal to our mutants (Zhang et al., "Coupled intra- and interdomain dynamics support domain cross-talk in Pin1", 2020, J. Biol. Chem., 295, 16585).

For FFpSPR, we do not observe the bending. This is also the case when we only use the eNOE set shared with the pCDC25 complex (where the same data set produces the bend).

6. line 375-382, page 12. Could the authors provide NOEs and information of backbone dynamics for the residues in the catalytic loop of Pin1 in the presence of pCDC25c and FFpSPR, so as to verify the rigidity of the catalytic loop in the two complexes?

We have recorded R_1 and $R_{1\rho}$ relaxation, and heteronuclear NOE data. We do not see significantly altered values for either complex. Due to figure number restriction, we prefer not to include these figures in the manuscript.

Figure 7. R_1 , R_2 (extracted from R_1 and $R_{1\rho}$), extracted overall tumbling times τ_C and heteronuclear NOE values versus residues numbers.

7. Could the authors design experiments to verify the “path 1” hypothesis after some residues in α -1 helix for FFpSPR binding are mutated?

We assume that the reviewer meant to write “path 2” rather than “path 1” as helix 1 is not part of “path 1”. Design of such a mutant is our plan and will be part of a grant application. We point out that this is a difficult endeavor: either, residues pointing into the PPlase core must be mutated that change the position of the helix or its dynamics, but do not thereby alter “path 1”. The second option is to find mutations that alter the interaction with the ligand bound to the WW domain. The challenge here is that we do not know the exact position of the ligand since no complex structures of that kind have been published. Our own NOEs between the ligands and Pin1 were not sufficiently conclusive (this probably explains why it has not been done in the past). If we succeed in designing a mutant that ONLY affects “path 2” (and this is hard to confirm), we will be able to quantify the contribution of “path 2” to the catalytic activity by conducting kEXSY experiments. Overall, we expect this to be a major project in its own.

8. A paragraph should be added to discuss why isomerase Pin1 has the flexibility in dealing with different ligands.

We agree with the reviewer that discussing Pin1’s ligand promiscuity helps put the project into context for biological implications and future directions.

Therefore, we have added paragraph at the end describing Pin1’s flexibility in different ligands:

“The conformational flexibility described here likely promotes the substrate interaction promiscuity in Pin1. Pin1 is known to have a large interactome; within cell cycle progression alone, Pin1 is known to interact with CDK1, CDC25c, Wee1, CENP-F, p53, and p27. Pin1 typically targets intrinsically disordered regions of substrate proteins, frequently involving multiple pS/T-P sites that vary in distance from 5 to 30+ residues. Some examples of multivalent Pin1 ligands include tau, full-length CDC25c, RNA Pol II, and IRAK1. Interdomain flexibility allows Pin1 to interact with a higher variety of multivalent ligands as proposed through a “fly-casting” model. On the other hand, there are also substrates that Pin1 only interacts with at one site, including b-catenin and NF κ B.”

REVIEWERS' COMMENTS

Reviewer #1 (Remarks to the Author):

Minor comment:

In the figures, when a 90 or 180 degree rotation is indicated, it would be helpful to include a curved arrow indicating the direction of rotation.

Reviewer #2 (Remarks to the Author):

The authors detail a number of revisions to the concerns raised in the first review. Important are the issues of to what degree differences in the proposed structural ensembles are real vs. a result of modeling differences, including different sets of restraints. While the revised manuscript provides some clarification of this, significant uncertainties remain as to the significance of the proposed structural changes. Difficulties remain in the overall lack of clarity with respect to the effect these changes have on the big picture. While the significantly improved manuscript should be published, it may be more appropriate for a more specialized journal.

Reviewer #3 (Remarks to the Author):

I have a few questions or comments:

(1) Ligands only transfer from WW domain to PPIase domain in the compact states of Pin1. Therefore, it looks like that the ID interfaces of Pin1 with different ligands like pCDC25c or FFpSPR may not be the same one, because Pin1 uses Path 2 for pCDC25c while it mainly uses Path 1 for FFpSPR.

Could the authors use one Figure of complex structures (or models) of Pin1 and pCDC25c/FFpSPR, in the compact states, to explain (by structures) why Pin1 chooses Path 2 for pCDC25c and Path 1 for FFpSPR? Such a figure may make the possible allosteric mechanisms more clear.

(2) "The degree of methyl occlusion from the interface with pCDC25c is likely dependent on the compact and extended states of the two-state ensemble" (lines 232-233), and others (lines 234-236). Could the

authors use one paragraph to discuss the possible function of coexistence of compact and extended states?

(3) Lines 291-293, it is difficult to understand the correlation “between the extended and compact states ...”

(4) Where are Fig 4d and 4e?

(5) In the whole text, when the authors describe “conformation change” of ligand-bound structure, is it referred to apo-form of Pin1 (for example, line 334)?

Reviewer #1:

□□ In the figures, when a 90 or 180 degree rotation is indicated, it would be helpful to include a curved arrow indicating the direction of rotation. □

We have added such arrows and the degrees of rotation to Figure 1, 3, and 4. □□

Reviewer #2 (Remarks to the Author): □□ The authors detail a number of revisions to the concerns raised in the first review. Important are the issues of to what degree differences in the proposed structural ensembles are real vs. a result of modeling differences, including different sets of restraints. While the revised manuscript provides some clarification of this, significant uncertainties remain as to the significance of the proposed structural changes. Difficulties remain in the overall lack of clarity with respect to the effect these changes have on the big picture. While the significantly improved manuscript should be published, it may be more appropriate for a more specialized journal. □

We believe that we provided clear evidence that the structural changes are driven by experimental restraints rather than by modeling differences.

□□ Reviewer #3 (Remarks to the Author): □□ (1) Ligands only transfer from WW domain to PPlase domain in the compact states of Pin1. Therefore, it looks like that the ID interfaces of Pin1 with different ligands like pCDC25c or FFpSPR may not be the same one, because Pin1 uses Path 2 for pCDC25c while it mainly uses Path 1 for FFpSPR. □ Could the authors use one Figure of complex structures (or models) of Pin1 and pCDC25c/FFpSPR, in the compact states, to explain (by structures) why Pin1 chooses Path 2 for pCDC25c and Path 1 for FFpSPR? Such a figure may make the possible allosteric mechanisms more clear.

Unfortunately, we do not know the exact positions of the ligands since our NOEs between the ligands and Pin1 were not sufficiently conclusive. In fact, no complex structure of that kind has been published at all, probably explaining our own failure. Therefore, we could only provide a high-resolution structure of the bound compact Pin1 state but not the ligand position.

□ (2) “The degree of methyl occlusion from the interface with pCDC25c is likely dependent on the compact and extended states of the two-state ensemble” (lines 232-233), and others (lines 234-236). Could the authors

use one paragraph to discuss the possible function of coexistence of compact and extended states? □

This is indeed an interesting question. A role of the coexistence of compact and extended states that immediately emerges from this study is a continuous rather than a binary control of the enzymatic activity of Pin1 through ligand binding to the WW domain. Although not investigated in this work, an additional role of the coexistence of these states may be a specific accommodation of ligands with multiple binding sites (for example, CDC25 has five binding sites). The function may be directly dependent on the separation of two such binding sites, which then finetunes the interaction with Pin1. Alternatively, this coexistence of states may provide a mechanism to bind two different molecules, each to one Pin1 domain, which in turn repositions the two bound molecules by a shift from extended to compact state of Pin1.

We already touch on this topic in the last paragraph in the Conclusion section of the previously submitted version. Therefore, we now integrate the aspects focusing on the states equilibrium into this paragraph:

“The conformational flexibility described here likely promotes the substrate interaction promiscuity in Pin1. Pin1 is known to have a large interactome; within cell cycle progression alone, Pin1 is known to interact with CDK1, CDC25c, Wee1, CENP-F, p53, and p27²²⁻²⁸. Some of the substrates interact with Pin1 only at one site, including β -catenin and NF κ B^{32,33}. For such cases, a role of the coexistence of compact and extended states that immediately emerges from this study is a continuous rather than a binary control of the enzymatic activity of Pin1 through ligand binding to the WW domain. Pin1 typically targets intrinsically disordered regions of substrate proteins, frequently involving multiple pS/T-P sites that vary in distance from 5 to 30+ residues²⁹. Some examples of multivalent Pin1 ligands include tau, full-length CDC25c, RNA Pol II, and IRAK1^{8,30}. Here, an additional role of the coexistence of these states may be a specific accommodation of ligands with multiple binding sites. The function may be directly dependent on the separation of two such binding sites, which then fine-tunes the interaction with Pin1. Alternatively, this coexistence of states may provide a mechanism to bind two different molecules, each to one Pin1 domain, which in turn repositions the two bound molecules by a shift from extended to compact state of Pin1. In these scenarios, interdomain flexibility allows

Pin1 to interact with a higher variety of multivalent ligands as proposed through a “fly-casting” model^{15,31}. We expect the detailed structures of ligand-bound Pin1 to be utilized to understand what peptide sequence determinants lead to differences in the interdomain equilibrium. Recent work has described the residues preceding pS/pT as most critical, with polar residues (as in peptide pCDC25c) stabilizing the extended state more than hydrophobic residues (FFpSPR)²⁰. Further exploration of substrate sequence determinants and the Pin1 in-cell interactome will aid in determining the functional consequences of Pin1’s malleable interdomain equilibrium.”

(3) Lines 291-293, it is difficult to understand the correlation “between the extended and compact states ...”

□ We rephrased the sentences. It reads now: “In addition, the extended and compact states feature distinct conformations around ...”

(4) Where are Fig 4d and 4e? □

This was a left over of the old version that we did not replace by mistake. We fixed this error.

(5) In the whole text, when the authors describe “conformation change” of ligand-bound structure, is it referred to apo-form of Pin1 (for example, line 334)?

We carefully checked the entire text. Generally, we use the expression “conformational change” in the context of ligand binding. We believe that this is clear from the context in all such cases. Examples are:

- in the title “**Ligand-specific conformational change**”
- In the abstract “Ligand binding..., but the conformational changes ” or “ligand-specific conformational changes ”
- In figure caption 1 we say “conformational changes due to ligand binding ”,
- Section title “*pCDC25c binding induces correlated intradomain conformational changes* ”

There is one exception, used in the Introduction section:

“cascades of conformational changes leading to allostery ”

Again, it is clear that this is in the context of any kind of allostery, be it ligand-, or temperature-induced, or any other kind of trigger.